# A Generative-Symbolic Model for Logical Reasoning in NLU

**Jidong Tian**[1,2] , **Yitian Li**[1,2] , **Wenqing Chen**[1,2] , **Hao He**[1,2*] and **Yaohui Jin**[1,2*]

[1]MoE Key Lab of Artificial Intelligence, AI Institute, Shanghai Jiao Tong University

[2]State Key Lab of Advanced Optical Communication System and Network, Shanghai Jiao Tong University

{frank92, yitian_li, wenqingchen, hehao, jinyh}@sjtu.edu.cn

## Abstract

Despite the recent success of language models (LMs) in natural language understanding (NLU), there are growing concerns about LMs' lack of logical reasoning abilities resulting in poor generalization and robustness. Facing these concerns, neuro-symbolic integration may be a solution, which guides neural networks to understand the whole reasoning process by symbolic reasoners. In this paper, we propose a generative-symbolic model to benefit the logical reasoning in natural language understanding. The experiment on the CLUTRR dataset shows that this neural-symbolic model performs better than the corresponding neural model.

## 1 Introduction

Language models (LMs), such as BERT [Devlin *et al.*, 2019] and RoBERTa [Liu *et al.*, 2019], have achieved success in many natural language understanding (NLU) tasks. However, there are growing concerns that these LMs cannot perform true reasoning based on logic [Ribeiro *et al.*, 2019; Sen and Saffari, 2020; Sugawara *et al.*, 2020]. An expected solution is the neuro-symbolic integration, which combines the extraordinary perception of neural networks and the stable reasoning ability of symbolic systems [Dai *et al.*, 2019]. However, it is still challenging for adopting available neuro-symbolic frameworks to NLU directly because of two reasons: 1) many NLU tasks cannot be described through simple rules, making it difficult to establish the rule system; 2) the symbolic system cannot backpropagate gradients to the neural network directly.

Inspired by [Li *et al.*, 2020], we propose a generative-symbolic model (GSM) to solve the logical reasoning problem defined on the CLUTRR dataset [Sinha *et al.*, 2019]. This dataset is a pure dataset to benchmark logical reasoning ability only in the field of kinship. Therefore, we can provide rule system with the simple form of $rel_1(x, y) \wedge rel_2(y, z) \rightarrow rel_3(x, z)$. For the example shown in Figure 1, the reasoning process can be described by a rule: $Mother(Carol, Kristin) \wedge Mother(Kristin, Justin) \rightarrow Grandmother(Carol, Justin)$.

---
[*]Corresponding Authors

**Context: Kristin** and her **son Justin** went to visit her **mother Carol** on a nice Sunday afternoon. They went out for a movie together and had a good time.
**Question:** How is **Carol** related to **Justin**?
**Answer:** Grandmother.

Figure 1: An example of the logical reasoning problem.

Based on such the rule system, our proposed GSM first includes a generative neural model to perceive information from contexts and build the reasoning process composed of a sequence of rules. According to pre-defined rules, GSM could make logical reasoning directly. To train the generative model without the supervision of the reasoning process, we adopt a Metropolis-Hastings sampler to sample pseudo labels to supervise the learning process [Li *et al.*, 2020]. The experiment shows that GSM has better generalization and robustness than end-to-end neural models as it can capture the correct logic to make the decision.

In conclusion, our contributions include:

- We propose a neuro-symbolic model, GSM, which can be adopted to the logical reasoning in NLU.

- The experiment on the CLUTRR dataset helps us to analyze the pros and cons of neural-symbolic integration in the field of natural language understanding.

## 2 Related Work

### 2.1 Logical Reasoning in NLU

Logical reasoning is one of the most widely used reasoning forms in NLU. Most classical natural language inference tasks, such as SNLI [Bowman *et al.*, 2015] and MNLI[Williams *et al.*, 2018], are strongly related to the logical reasoning form. With the great success of LMs on these datasets, logical reasoning is considered a solved problem. However, some recent question-answering studies have revealed that state-of-the-art LMs still lack the logical reasoning ability. Evaluations on LogiQA [Liu *et al.*, 2020] and ReClor [Yu *et al.*, 2020] have exposed that such models cannot truly understand the logical rules and perform deduction. Meanwhile, experiments on CLUTRR [Sinha *et al.*, 2019] show neural models' lack of robustness and generalization ability when making logical reasoning, as

they tend to capture spurious correlations rather than understand the reasoning process [Kaushik and Lipton, 2018; Sugawara *et al.*, 2020].

## 2.2 Neuro-Symbolic Integration

Neuro-symbolic integration is a concept that combines connectionism and symbolism to make machines have better perception and reasoning abilities [Kim *et al.*, 2020]. Implicit integration methods include constructing logical neural architectures [Saralajew *et al.*, 2019; Chen *et al.*, 2020] and mapping symbolic rules to the feature spaces [Xu *et al.*, 2018; Li and Srikumar, 2019]. These methods provide logical constraints, but they are not fully capable of supervising models learning logical reasoning. Explicit methods are to complete the perception and reasoning processes in two spaces and interact through specific methods, such as logic programming [Dai *et al.*, 2019] and reinforcement learning [Liang *et al.*, 2018].

## 3 Problem Setting

In the logical reasoning problem, the final objective is to find a model $F$ that maps contexts $x$ and a set of pre-defined rules $R$ to the ground truth $y$ [Dai *et al.*, 2019]. Therefore, the expected model should satisfy the condition in Eq 1.

$$\forall (x, y) \in D(F(x, R) \models y) \tag{1}$$

However, it is hard to directly find a neural model $F$ that can effectively model $x \cup R \to y$ because the reasoning process with $R$ is non-differentiable. Neuro-symbolic integration provides a novel solution to solve the problem by introducing a mediator $z$ to decouple the model $F$ to a neural model $nn$ and a symbolic reasoner (represented by the ruleset $R$). Therefore, the Eq 1 can be rewritten as Eq 2, where $\models$ stands for logical entailment.

$$\forall (x, y) \in D(z = nn(x, R), z \cup R \models y) \tag{2}$$

## 4 Methodology

Based on the above definition, GSM includes the perceptual neural network, symbolic reasoner (a set of rules $R$), and the training module that end-to-end train the whole models.

### 4.1 Neural Encoder

Based on the analysis on CLUTRR, $z$ can be defined as the reasoning process consisting of a sequence of rules. Therefore, the GSM adopts an LSTM-based encoder-decoder framework to generate $z$. To compare with the neural networks, we take three baselines of CLUTRR (BiLSTM, fixed BERT, and fixed RoBERTa) as the encoder and a modified LSTM-based decoder.

### 4.2 Semi-Supervised Training Module

Based on the Eq 2, the final objective is to maximize the probability $p_\theta(y|x, R)$, where $\theta$ represents parameters to be optimized. To calculate the objective probability by the mediator $z$, we can marginalize over $z$ based on the independent condition $(x \perp y|z)$, shown in Eq 3, where $p(y|z, R)$ is generated by $R$ without parameters, and $p_\theta(z|x, R)$ is generated by the neural network with parameters $\theta$.

$$\begin{aligned} p(y|x, R) &= \sum_z p(y, z|x, R) \\ &= \sum_z p(y|z, R)p_\theta(z|x, R) \quad (x \perp y|z) \end{aligned} \tag{3}$$

Based on the maximum likelihood estimation, the derivation of the likelihood is shown in Eq 4.

$$\begin{aligned} \nabla_\theta L &= \nabla_\theta \log p(y|x, R) = \frac{\nabla_\theta p(y|x, R)}{p(y|x, R)} \\ &= \sum_z p(z|x, y, R) \nabla_\theta \log p_\theta(z|x, R) \\ &= \mathbb{E}_{z \sim p(z|x,y,R)}[\nabla_\theta \log p_\theta(z|x, R)] \end{aligned} \tag{4}$$

Based on Eq 4, the key point is to model the objective distribution of $z \sim p(z|x, y, R)$. Actually, this posterior distribution can be simplified as Eq 5, because the effective reasoning processes should match the rules $R$ and model $y$. To avoid the computational complexity of the summation $\sum_{z \in \{z|z \cup R \models y\}} p_\theta(z|x)$, an alternative solution is to sample $z$ from the posterior distribution $p(z|x, y, R)$ through Metropolis-Hastings (M-H) sampler.

$$p(z|x, y, R) = \begin{cases} 0, & , z \cup R \nvDash y \\ \dfrac{p_\theta(z|x)}{\sum_{z'} p_\theta(z'|x)}, & z \cup R \models y \end{cases} \tag{5}$$

**Metropolis-Hastings Sampler.** To achieve M-H sampling, we regard $\pi(z) = \frac{p(z|x,R)}{\sum_{z'} p(z'|x,R)}$ as the desired stationary distribution, and design a distribution $Q$ as the proposal distribution. Therefore, when given the current state $z_i$ and the sampled state $z_j$, the acceptance ratio can be calculated by Eq 6, and $p(z|x, R)$ is the probability calculated by the outputs of the neural network. To ensure the convergence of the desired distribution, the sampling process will iterate $N$ times. As the neural network is difficult to generate the correct $z$ once, we always accept the sample for the first iteration. The algorithm is shown in Alg 1. Based such a sampler, we will introduce how to sample $s$ from $Q$ and calculate the probability $p_{nn}$ and the acceptance ratio $\alpha$.

$$\begin{aligned} \alpha(z_i, z_j) &= \min\{\frac{\pi(z_j)Q(z_i|z_j)}{\pi(z_i)Q(z_j|z_i)}, 1\} \\ &= \begin{cases} 1, & z_i \cup R \nvDash y \\ \min\{\dfrac{p(z_j|x, R)Q(z_i|z_j)}{p(z_i|x, R)Q(z_j|z_i)}, 1\}, & z_i \cup R \models y \end{cases} \end{aligned} \tag{6}$$

**One-Step Sampler.** To sample $z$, we introduce a search method. As mentioned above, each $z$ consists of a sequence of rules. when given the rule's head ($rel_3$ of $rel_1 \wedge rel_2 \to rel_3$ in CLUTRR), we can randomly search the rule's body ($rel_1$ and $rel_2$) based on $R$. Therefore, the one-step sampling can be executed by back adjusting the rule sequence from the end to the beginning. Based on such the sampler, although

**Algorithm 1:** M-H Sampler.

**Input:** Probability $p_{nn}(\cdot|x, R)$; Step $N$; $Q(\cdot|z)$.
**Output:** Sampled $z^*$

1   Sample $z^{(0)} \sim p_{nn}(\cdot|x, R)$;
2   **for** $t = 1$ to $N$ **do**
3     Sample $z \sim Q(\cdot|z^{(t)})$;
4     **if** $t == 1$ **then**
5       $z^{(t+1)} = z$
6     **else**
7       Sample $u \sim \mathcal{U}(0, 1)$;
8       $\alpha = \min\{\frac{p_{nn}(z|x,R)Q(z^{(t)}|z)}{p_{nn}(z^{(t)}|x,R)Q(z|z^{(t)})}, 1\}$;
9       **if** $u \leq \alpha$ **then**
10        $z^{(t+1)} = z$
11       **else**
12        $z^{(t+1)} = z^{(t)}$
13       **end**
14     **end**
15   **end**
16   **return** $z^* = z^{(N)}$

$Q$ cannot be given, $\frac{Q(z_i|z_j)}{Q(z_j|z_i)}$ can be calculated directly. When given a rule's head $r$, we define a function $G(r, R)$ to represent how many rules in $R$ will imply $r$. For two mediators $z_i$ and $z_j$, their sequences of rule's head can be extracted as $r_i^k$ and $r_j^k$. Then $\frac{Q(z_i|z_j)}{Q(z_j|z_i)}$ can be calculated by Eq 7.

$$\frac{Q(z_i|z_j)}{Q(z_j|z_i)} = \frac{\prod_{r \in \{r_j^k|r_j^k \neq r_i^k\}} G(r, R)}{\prod_{r' \in \{r_i^k|r_i^k \neq r_j^k\}} G(r', R)} \quad (7)$$

### 4.3 Loss Function

The sampled $z^*$ is regarded as the pseudo label to train GSM. The loss function can be computed by the cross-entropy loss $L = \text{CE}(z, z^*)$. Based on this method, our neuro-symbolic network can be optimized in a semi-supervised manner that only takes advantage of the final label $y$ and logical rules $R$.

## 5 Experiment

### 5.1 Dataset and Baselines

We experiment on CLUTRR [Sinha *et al.*, 2019], which is a logical benchmark dataset [Gontier *et al.*, 2020]. The original dataset provides contexts (including stories and questions) and labels (kinships) as input-output pairs $(x, y)$. The logical reasoning process $z$ can be simplified as a sequence of rules of kinships, such as $Grandmother : -Mother, Mother$. To evaluate GSM, we select two baselines from [Sinha *et al.*, 2019], Bi-LSTM, and BERT as encoders to establish GSM. Although GAT [Velickovic *et al.*, ] and CTP [Minervini *et al.*, 2020] can perform better on CLUTRR, they attempts learn from the structural triplets but not from the raw texts. Besides, DeepProbLog [Manhaeve *et al.*, 2018] also provides the baseline for CLUTRR, it adopts different experimental settings from the original work [Manhaeve *et al.*, 2021].

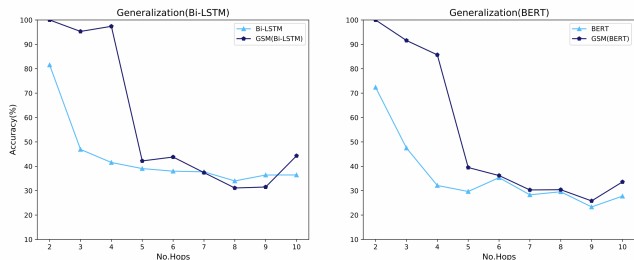

Figure 2: Generalization results of Bi-LSTM, BERT, and their corresponding GSMs.

### 5.2 Metrics

Generalization and robustness are two metrics defined on CLUTRR [Sinha *et al.*, 2019]. Generalization is the metric to measure how models generalize to more-hop reasoning problems. We train models on 2, 3, and 4-hop instances and test on an up-to-10-hop set. Robustness is the metric to how models perform in different scenarios. CLUTRR provides four kinds of datasets: (1) Clean Data; (2) Supporting Data containing contexts with evidence to support the answer; (3) Irrelevant Data whose contexts provide noisy expressions irrelevant to the answer; (4) Disconnected Data with contexts that provide noisy expressions disconnected with the query entities. In reality, irrelevant datasets contain more information than clean sets, while the latter two datasets provide more noise.

### 5.3 Hyper-Parameters

Hyper-parameters are shown in Table 1. For generalization, we adopt the hyper-parameters from the original experiments on CLUTRR [Sinha *et al.*, 2019]. For robustness, we seach the learning rate from $[1e-6, 1e-5, 1e-4, 1e-3]$, while we search L2 coefficient from $[1e-3, 1e-4, 1e-5]$.

| Parameter | Generalization | Robustness |
|---|---|---|
| LR | 1e-3 | 1e-5 |
| L2 | 0 | 1e-3 |
| Dropout | 0 | 0.5 |
| LR Decay | 0 | 0 |
| Epochs | 100 | 100 |
| ES | 20 | 50 |
| Optimizer | ADAM | ADAM |

Table 1: Hyper-parameters. LR: learning rate; L2: L2 coefficient; ES: early stop.

### 5.4 Results of Generalization

Results of generalization are shown in Figure 2. From Figure 2, GSM performs better than their corresponding baselines on in-domain sets (up to 4 hops). However, GSM not always performs better than Bi-LSTM, especially on higher-hop reasoning problems (7 to 9-hop reasoning). Based on our analysis, one of the reasons is that the neural generator cannot effectively determine when to stop reasoning. Nevertheless, the overall performance of GSM is better than the two baselines' because GSM introduces the reasoning process to avoid the spurious reasoning process.

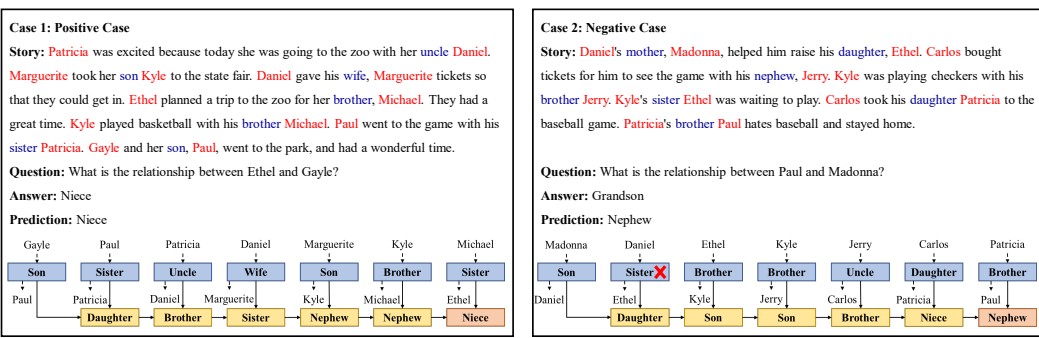

Figure 3: Case study with a positive case and a negative case.

| Train | Test | Bi-LSTM | | BERT | |
|---|---|---|---|---|---|
| | | Base | GSM | Base | GSM |
| Clean | Clean | **99.1** | 68.5 | 29.6 | 33.3 |
| | Supporting | 45.5 | **66.3** | 26.7 | 26.7 |
| | Irrelevant | 49.0 | **77.0** | 29.0 | 44.0 |
| | Disconnected | 21.0 | **36.0** | 28.0 | 30.0 |
| Supporting | Clean | **67.3** | 53.5 | 19.8 | 28.7 |
| | Supporting | **67.5** | 48.6 | 32.5 | 43.9 |
| | Irrelevant | **66.6** | 47.4 | 18.8 | 24.8 |
| | Disconnected | 48.8 | **51.0** | 24.2 | 18.0 |
| Irrelevant | Clean | 51.6 | **63.0** | 14.1 | 28.0 |
| | Supporting | 55.5 | **64.7** | 28.9 | 19.0 |
| | Irrelevant | 51.8 | 62.9 | 15.3 | **68.6** |
| | Disconnected | 42.0 | **68.3** | 23.3 | 28.7 |
| Disconnected | Clean | 35.2 | **63.0** | 13.3 | 56.0 |
| | Supporting | 53.1 | **67.0** | 21.9 | 40.0 |
| | Irrelevant | 43.1 | **53.5** | 27.2 | 29.7 |
| | Disconnected | 33.3 | 52.8 | 26.1 | **57.5** |

Table 2: Robustness results.

## 5.5 Results of Robustness

Table 2 shows the results of robustness experiments. In most cases, the performance of GSM on the out-of-domain evaluation is better than its corresponding baseline model, except Bi-LSTM trained in the supporting scenarios. As mentioned above, the supporting datasets provide more information than the other three kinds of datasets, resulting in better robustness of end-to-end Bi-LSTM. This means that end-to-end neural models benefit from more evidence provided by the datasets. On the contrary, GSM is more stable and robust than end-to-end baselines when trained in the different scenarios as it introduces the reasoning process to constrain the reasoning result. This is evidence that the proposed reasoning process benefits organizing relevant information and avoiding interference from irrelevant information.

## 5.6 Case Study

Based on the analysis of generalization and robustness, we will further discuss the interpretability of GSM through two typical cases. Figure 3 shows a positive case (Left) and a negative one (right). The positive case shows that GSM can completely capture the true reasoning process, which illustrates its interpretability. Although the negative case reflects GSM cannot make the correct predictions to some degree, we can also trace back why GSM mispredicts the "grandson" as "nephew". In reality, GSM fails to perceive both two kinships in "Daniel's mother, Madonna, helped him raise his daughter, Ethel", so that GSM incorrectly identifies the "daughter" as the "sister". Although the subsequent reasoning is correct, the whole reasoning process is invalid. This case reflects a feature of GSM that GSM has strong reasoning ability but poor perceptual ability because the supervision signals concentrate more on the reasoning process.

## 5.7 Limitations

Based on the above analysis, the biggest challenge for GSM is that it can only effectively constrain the reasoning process but cannot effectively control the perceptual process, as there are no ground-truth labels for the mediator's generation. We also summarize other limitations of GSM below.

- As a result of the lack of supervision, GSM cannot quickly converge in the cold start setting. In reality, it will bring further improvements when adopting neural models with better perceptual abilities.

- It is challenging to automatically stop the generative process, which leads to relatively insignificant improvements in the generalization experiment.

- GSM cannot automatically distinguish different types of reasoning forms, making it hard to apply in scenarios where multiple reasoning forms are coupled in the context.

## 6 Conclusion

In this paper, we propose a neuro-symbolic model, GSM, to generate the reasoning process in a semi-supervised manner. Results on CLUTRR show that GSM has better generalization and robustness than end-to-end neural models, although GSM has many limitations that hinder its application in more general language scenarios. Also, this work motivates future studies on 1) more general paradigms to describe diverse reasoning forms in NLU tasks; 2) better neural encoders to enhance the models' perception ability; 3) more human-like neuro-symbolic systems to complete complex NLU tasks.

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
