# OpenReview forum: "A Generative-Symbolic Model for Logical Reasoning in NLU"
_ijcai.org/IJCAI/2021/Workshop/NSNLI — NSNLI Oral_

### Official Review · Reviewer_oTWj · 2021-05-19
**Review for A Generative-Symbolic Model for Logical Reasoning in NLU: relevant for the workshop, but needs a better comparison to related work.**

**Rating:** 6
**Confidence:** 5

**Review:**

The authors introduce a new neural-symbolic method GSM that integrates language models with rule-based reasoning. The method is centered around sampling pseudo-labels for the neural model that extracts facts from the natural language that is used in the reasoning method.  The authors evaluate their approach on the CLUTRR dataset, and show an increase in performance over a fully neural baseline. This makes the paper relevant for the workshop.

To further strengthen the paper, the authors might want to consider the following points:
- Currently, it is somewhat unclear what rules are exactly given, and what part of the rules (if any) should be learned.
- Several other neural-symbolic papers have performed very related work on the CLUTRR dataset. Examples include:

1) Minervini, P., Riedel, S., Stenetorp, P., Grefenstette, E. & Rocktäschel, T.. (2020). Learning Reasoning Strategies in End-to-End Differentiable Proving. Proceedings of the 37th International Conference on Machine Learning, in Proceedings of Machine Learning Research 119:6938-6949 Available from http://proceedings.mlr.press/v119/minervini20a.html .

2) Manhaeve, R., Dumancic, S., Kimmig, A., Demeester, T., De Raedt, L. (2021). Neural probabilistic logic programming in DeepProbLog. Artificial Intelligence, 298, Art.No. 103504. doi: 10.1016/j.artint.2021.103504

---

### Official Review · Reviewer_9cN8 · 2021-05-25
**Improve Algorithm Presentation, Add Comparison to Related Work**

**Rating:** 6
**Confidence:** 2

**Review:**

The authors introduce an approach to train neural models with an additionally available set of rules. Since rules themselves are non-differentiable, they come up with a method to train the network using the rules by introducing a mediator variable z. The approach involves sampling the variable z* using distribution learnt by the neural network and the rules and then training the model against the predicted value and the sample z*.  The proposed approach effectively combines rule based reasoning to improve neural network prediction performance and thus is a good fit for the workshop.

Some suggestions to improve:
1. There are some typos in the paper:  ... perform truly reasoning based on logic ..., ... acceptance ration ...
2. While the authors have presented the MH Sampler algorithm in pseudocode. It might be good to present the whole training process via some diagram to make it clearer.
3. There is no comparison with any other baseline and there is no explanation given as to why other works have not been compared against.

---

### Decision · Program_Chairs · 2021-05-27

**Decision:**

Accept (Oral)

**Comment:**

The paper tackles the topic clearly relevant for the workshop.
Please take into account the comments of the reviewers when preparing the camera-ready versions.